# Study protocol: The role of milk matrix lipids in programming the immunoreactivity of proteins derived from lactic acid bacteria

**Anna Maria Ogrodowczyk**[1]*, **Ewa Romaszko**[2]

1 Polish Academy of Sciences, Institute of Animal Reproduction and Food Research, Olsztyn, Poland,
2 Non-Public Health Care Clinic "ATARAX", Olsztyn, Poland

* a.ogrodowczyk@pan.olsztyn.pl

## Abstract

Food allergy is widely recognized as a significant health issue, having escalated into a global epidemic, subsequently giving rise to the development of numerous additional complications. Currently, the sole efficient method to curb the progression of allergy is through the implementation of an elimination diet. The increasing number of newly identified allergens makes it harder to completely remove or avoid them effectively. The immunoreactivity of proteins of bacterial origin remains an unexplored topic. Despite the substantial consumption of microbial proteins in our diets, the immunologic mechanisms they might induce require thorough validation. This stands as the primary objective of this study. The primary objective of this study was to evaluate the effects of bacterial proteins on the intestinal barrier and immune system parameters during hypersensitivity induction in both developing and mature organisms. The secondary objective was to evaluate the role of lipids in the immunoreactivity programming of these bacterial proteins. Notably, in this complex, comprehensively designed *in vitro*, *in vivo*, and *ex vivo* trial, the immunoreactivity of various bacterial proteins will be examined. In summary, the proposed study intends to address the knowledge gaps regarding the effects of *Lactobacillus* microbial proteins on inflammation, apoptosis, autophagy, and intestinal barrier integrity in a single study.

## Introduction

In the EU, the issue of food allergies and their associated complications is on the constant rise, affecting about 11–26 million people yearly [1, 2]. As a consequence, the global expenses related to allergy therapy and diagnostics are increasing (2.69 billion USD in 2018, with a predicted increase of 7.4% by 2028) [3]. While an elimination diet remains the sole effective way to curb allergy progression, it significantly influences various health aspects of patients' lives and causes discomfort. Cow milk proteins, as the initial source of cross-species protein, are considered to be the primary food-origin antigens that affect an infant's development. However, proteins from microorganisms present in food, including dairy products, have not received adequate attention in this context. Notably, the list of known antigens has expanded

relevant data from this study will be made available upon study completion.

**Funding:** A.M.O. This study was financially supported by the National Science Centre, Poland (Project No. 2021/43/D/NZ9/02814). URL: https://projekty.ncn.gov.pl/en/index.php?projekt_id=536724 The funders did not and will not have a role in study design, data collection and analysis, decision to publish, or preparation of the manuscript.

**Competing interests:** The authors have declared that no competing interests exist.

to include yeast proteins with proven sensitizing potential (i.e., enolase and peroxisomal protein) [4]. Additionally, it was also reported that certain yeast heat shock proteins (Hsp) possess the potential to modulate immune responses in developing newborn organisms [5]. In this context, it is plausible that not only milk proteins but also microbial, including bacterial, proteins may contribute to the initial hypersensitivity of the body and play a role in the progression of allergies. Therefore, effective elimination may not be always possible due to the ubiquitous distribution of microorganisms across various niches, including food.

In many protocols dedicated to allergy therapy, probiotics and fermented dairy products (FDP) containing bacteria and yeast are used as immunomodulatory agents [6]. Preparations containing lactic acid bacteria (LAB) are also used as a protective measure and in the prophylaxis of other hypersensitivity-related diseases. Various bacterial structures and metabolites (i.e., exopolysaccharides, volatile fatty acids, and nucleic acids) have demonstrated their ability to modulate the immune system effectively [4, 7]. However, it is important to highlight that dairy beverages and probiotic preparations often contain high numbers of microbial cells. Overlooking the microbial protein content in dairy products is a common oversight.

It has been estimated that the mentioned expression of yeast proteins reaches approximately 6,000 different proteins, with analyses revealing that each cell may produce around 42 million protein particles [4]. Bacterial cells also contain a significant number of proteins in their structure, serving various functions and replication potential and exhibiting a relatively high degree of homology to yeast proteins (e.g., enolase). On average, each live bacterial cell can produce about 15 femtograms of conserved bacterial proteins (BPs) [8]. Importantly, protein expression is intricately linked to taxonomic affiliation as well as the growth conditions of bacteria. In dairy technology, the proper interaction of the bacterial cultures with the raw materials plays an essential role in the process efficiency and ensuring product safety. In addition, FDP are essential components of a balanced diet, and their safe consumption is of paramount importance. It also seems crucial to evaluate the immunoreactivity of the BPs they produce, particularly given the high adaptability of microorganisms. Unfortunately, traditional fermentation cultures with limited adaptive capacity are still used in the pursuit of improved biological activities of FDP when fermenting alternative and demanding matrices (i.e., sour whey, milk of exotic mammals, alternative protein sources, or plant-based drinks). The selection of raw materials and the appropriate strains with enzymatic activity tailored for fermentation seems to be crucial for the preparation of safe fermented products. Research has demonstrated that even slight variations in the final pH during the fermentation process (pH 4.8–5.8) can lead to significantly diverse expression profiles of up to 92 proteins in *Lactobacillus rhamnosus* GG, affecting their glycolytic enzyme phosphorylation [18]. Furthermore, *Lactobacillus casei* GCRL163 (new nomenclature: *Lacticaseibacillus casei* GCRL163), cultured under conditions of lactose starvation, demonstrated differences in the expression of eleven glycolytic enzymes, potentially representing a survival strategy under challenging growth conditions [9]. However, the immunoreactivity of those proteins has yet to be investigated. Recently, certain cell shock factors with a persistent acidity have been suggested to upregulate the expression of the S protein of *Lactobacillus paracasei* GCRL46 (n.n. *Lacticaseibacillus paracasei*. GCRL46). This protein shares homology with proteins characteristic for pathogenic Streptococci [10, 11]. Unfortunately, despite the selectivity of the human reaction, a defensive response toward the S protein of *L. paracasei*, a bacterium considered to be part of the normal human microbiota, has been observed.

The majority of identified microbial and reported proteins are recognized as moonlighting proteins. Although they are quite prevalent in bacteria over the past 25 years, their role in immune system regulation needs further investigation [12–14].

In light of the mentioned studies, it is not surprising that the consumption of probiotic preparations has, in some cases, led to a deterioration in the health of individual patients suffering from allergies and atopy [15, 16]. However, it is important to note that the consumption of probiotics and FDP supports the natural intestinal microbiota and helps alleviate the hypersensitization processes that support convalescence in various diseases [7, 17]. There are exceptions to this, particularly in cases where people experience the disruption of the MyD88-ROR-γt regulatory axis, often associated with dysbiosis resulting from factors like food allergies [18]. This axis plays a crucial role in restoring immune tolerance in FA, and its disruption results in excessive IgE-mediated reactions, including reactions to proteins of the host's own physiological gut microbiota. The ongoing research aims at enhancing the safety of FDP, probiotics, and synbiotics for all consumers, including those who may have compromised innate immunity but are unaware of it. Ensuring food safety is a fundamental aspect of nutrition and food research.

Interestingly, the consumption of FDP has experienced significant growth on a national scale, averaging a 42% increase over the past decade. However, this growth seems to be slowing down and entering a plateau phase. Moreover, the global forecasts made for the period from 2019 to 2029 indicate a decline in the consumption of FDP [19]. Among factors that constrain consumption growth, consumers often cite their subjective perception of lower digestibility and tolerance to FDP. The ~~applicant's~~ author's preliminary studies indicate that lactic acid BPs may be subjected to similar principles that govern intestinal microbiota proteins [20]. Although healthy organisms typically respond appropriately to consumed BPs, these proteins could potentially act as antigens for organisms with an imbalanced Th1/Th2 ratio and defect in the MyD88-ROR-γt axis. An emerging question that needs addressing is how to transition from a state of sensitivity to BPs to one of tolerance or how to reduce the IgE-immunoreactive potential of lactic acid bacteria (LAB) proteins.

In the author's preliminary studies, it was observed that some LAB strains, when cultured comparatively in sweet buttermilk and milk (0.5% or 3.2% fat), expressed BPs that exhibited varying levels of immunoreactivity intensities with antibody classes E. One of these proteins, cyclopropane-fatty-acyl-phospholipid synthase, was found to be IgE-immunoreactive, but its expression in *L.casei* LcY cell membrane varied depending on the type of raw material used (the intensive expression was observed in buttermilk matrix but not in cows' milk containing 3.2% fat). These findings led to the conclusion that lipids in the dairy matrix played a dual role in influencing the immunoreactivity of BPs. First, the different lipid compositions (i.e., especially phospholipids, trace lysophospholipids, and triglycerides) lead to the expression of different profiles of BPs (including constitutive proteins shock proteins and lipid metabolism-involved proteins). Second, lipids exhibit varying degrees of effectiveness in protecting bacterial cells from stressors including low pH, lactose deficiency, digestive system enzymes, and bile salt impact that causes the differences in Hsp BPs profile. Therefore, there is a pressing need for comprehensive investigations on the effects of lipids in LAB BPs profile modulation. Górska et al. [21] examined the IgE-immunoreactivity of several proteins of *Bifidobacterium* sp. in their studies, primarily focusing on the protein identification and characteristics rather than the regulation of their expression. Similarly, limited studies exist exploring the mechanisms of action of LAB and commensals BPs affect the immune system [22, 23]. Most of those studies predominantly report on the direct or indirect impacts of immunoreactive proteins from pathogens and potential pathogens like *Streptococcus pneumoniae* or *Escherichia coli* on the immune system [10, 11]. BPs are believed to play an important role in cell signaling, bacterial cell adhesion to the epithelial barrier, and immune response regulation. BPs have been shown to interact with various proteins involved in immune regulation, such as toll-like receptors (TLRs) and inflammasome elements, and it has been demonstrated that they may function

as a danger-associated molecular pattern (DAMP), stimulating immune responses by activating pattern recognition receptors (PRRs) on immune cells [24]. However, there are few reports on this topic regarding the role of LAB and commensal bacteria proteins. The planned experiment aims to verify the existing reports and conduct *in vitro* trial tests using epithelial cell lines and female BALB/cJRj mice as *in vivo* and *ex vivo* models. This will enable the exploration of the mechanisms underlying BPs immunoreactivity.

## Research objective and hypothesis

Building upon the insights presented in the previous section, the proposed study has been crafted to ascertain the immunogenicity of lactic acid BPs from cultures modified with lipids. Drawing on our current knowledge and the findings of our pilot study [25, 26], we hypothesize that a distinct profile of LAB proteins, induced in response to the lipids of the milk matrix, may serve to modulate the mechanisms linked to type I hypersensitivity and exert a beneficial impact on the development and sealing of the intestinal barrier.

The primary objective of these experiments was to establish the predominant role of milk matrix lipids in programming the immunoreactivity of proteins derived from lactic acid bacteria. The goal will be achieved using the following three specific aims:

Specific Aim 1: To detect, identify, and conduct a comparative profiling of the immunoreactive bacterial proteins of three selected *Lactobacillus* strains. The comparison of immunoreactive BPs profile isolated from bacteria cultured will be performed in the layout BPs isolated from optimal medium *vs.* proteins isolated from bacteria cultured in matrix fortified with the defined lipids composition. These assessments will be conducted using human atopic sera.

Specific Aim 2: To evaluate the immunoreactive/immunomodulatory properties of BPs *in vitro* studies. This will involve (a) investigating their impact on Caco- 2 and HIEC-6 human intestinal epithelial cell lines at various stages of maturity under different conditions, (b) assessing their effects in the presence or absence of induced inflammation, and (c) in monoculture and coculture with PBMCs.

Specific Aim 3: To assess the immunoreactive/immunomodulatory properties of BPs *in vivo* and *ex vivo* studies. It will be performed in a mouse model (BALB/cJRj) and *ex vivo* with cultures of induced mice lymphocytes (primary cultures).

The targeted selection of raw material with an optimal lipid composition significantly enhances the safety and nutraceutical properties of fermented products produced using bacteria in their manufacturing process. The anticipated outcomes of basic research hold the potential to provide crucial insights into the involvement of immunoreactive BPs in the development of allergies and other diseases caused by hypersensitivity. This, in turn, may lay the foundation for implementing the analysis of bacterial immunoreactivity as a novel discriminant of food safety. Furthermore, functional analysis of BPs and its impact on the mechanisms occurring in intestinal and immune cells will allow us to broaden the general knowledge about their putative role in the course of the allergic/tolerance development process.

## Experimental design

The immunoreactivity of BPs isolated from *L. casei* LcY, *L. bulgaricus* 151, and *L. rhamnosus* GG will be tested. These species are the most common in fermented food processing technologies and probiotic preparations. Strains will be cultured in optimal Man, Rogosa, Sharpe (MRS) medium without additives (control) or with different lipids fortification (treated

**Table 1. Concentration of lipids in fortified media.**

| Modified MRS media | TG | Total PL | PC | PE | PI | PS | SM | LPC | LPE |
|---|---|---|---|---|---|---|---|---|---|
| | % | mg/g fat | PL composition, expressed as % of the total PL | | | | | | |
| CM | 3.2 | 0.6 | 26.8 | 25.8 | 14.0 | 1.5 | 26.8 | 0.1 | 0.1 |
| BM | 0.5 | 4.49 | 27.0 | 25.7 | 5.8 | 9.7 | 20.4 | 0.7 | 1.0 |

CM, cow's milk; BM, buttermilk; TG, triglycerides; PL, phospholipids; PC, phosphatidylcholine; PE, phosphatidylethanolamine; PI, phosphatidylinositol; PS, phosphatidylserine; SM, sphingomyelin; LPC, lysophosphatidylcholine; and LPE, lysophosphatidylethanolamine.

groups). The impact of varying triglycerides (TG) and phospholipids (PL) (i.e., PC—phosphatidylcholine; PE—phosphatidylethanolamine; PI—phosphatidylinositol; PS—phosphatidylserine; SM—sphingomyelin and its trace variants (TPL); LPC—lysophosphatidylcholine; and LPE—lysophosphatidylethanolamine) on BPs expression will be examined through a comparative proteomic analysis. Tested lipid compositions will mimic the proportions found in different dairy matrices: (Table 1): full fatty cow's milk (CM; 3.2% TG, 0.3% PL, and 0.1% TPL), unfermented buttermilk (BM; 0.5% TG, 1.8% PL, and 0.3% TPL) [27]. These compositions will serve as model conditions that reflect different dairy raw materials while excluding additional factors like different contents of lactose, vitamins, and modulating factors.

## Immunoreactivity determination

The first experiment, aimed at achieving the primary goal, will follow the scheme outlined in Fig 1. Cultures will be established using a 3-fold passage system at 2-week intervals, allowing the bacteria to adapt to the culture conditions. We will assess bacterial viability and metabolic activity using a combination of both conventional microbial culturing methods and molecular analysis techniques, including gene expression analyses and cytometric characteristics of cell viability and metabolic activity. Throughout the experiment, we will continuously monitor the concentration of lipids in the medium and their degree of oxidation [28]. BPs will be isolated for further analysis. BPs fractions will be extracted using an optimized by authors' department protocol developed by Klaasens et al. [26, 29]. These isolates will undergo dialysis (from EPSs), filtration (0.2 μm-PVDF filters), and LCMS comparative proteomic characteristics. Additionally, we will evaluate the immunoreactivity of these BPs using pooled human, polyallergic sera. Sera of $n$ = 16 polyallergic to food patients (FA), and an equal number of healthy controls will

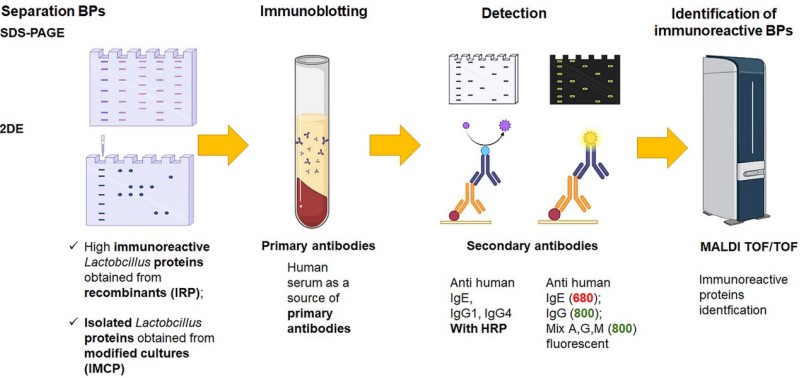

**Fig 1. Scheme of *in vitro* study with the IgE-immunoblotting with human sera.**

be used. The number of required sera was determined based on the experience in preliminary studies, and calculated using statistical tests for the minimal group quantity. These sera have already been collected and described [30, 31] in the course of the completed project [N N312311939]. Polyallergic sera (to cows' milk, whey, peanut, soy, and eggs—major allergens in the Polish population and atopic sera) will be used to encompass a broad spectrum of specific IgE and attain a high content of total IgE for testing. The proposed approach (not to target a specific allergy) is deliberate. Our previous research [26] has shown that the reaction to BPs is associated with a generally induced type I hypersensitivity to food, characterized by IgE secretion rather than a specific allergy. We employ a two-color fluorescent labeling method, developed in the authors' department to enhance the sensitivity and precision of BPs analysis [32], to detect BPs IgE-reactive, which will be finally identified with MALDI-TOF and MS/MS. Identified IgE-immunoreactive BPs secretion (i.e., GroEL, GAPDH, LDH, and enolase) will be monitored among others in bacterial cultures. Recombinant IgE-immunoreactive BPs (IRP) will be procured and used as sensitizing proteins for the stimulation of cell cultures and immunizing mice, in the second and third experiments, respectively. Isolated *Lactobacillus sp*. proteins obtained from modified lipids cultures (IMCP) will be tested as treatment variants.

## Impact of BPs on epithelial barrier integrity and physiology in *in vitro* study

The second experiment aims to investigate the effects of IRP and IMCP on the process of mucosal differentiation and maturation, with a focus on their impact on epithelial barrier integrity modulation. The evaluation of immunoreactive/immunomodulatory properties of IRP and IMCP will be conducted *in vitro*, utilizing (a) Caco-2 and HIEC6 human intestinal epithelial cell lines at various stages of maturity, (b) as well as in different conditions with and without induced inflammation, and (c) in monoculture and coculture with PBMC. The scheme of experiment 2 is shown in Fig 2. This experiment holds particular significance as sensitization often commences in the early stage of life when the maturation of the epithelial barrier is not completed yet. In experiment 2, HIEC6 (Fetal origin Intestine Epithelial cell line) will be used. The monitoring of the differentiation and maturation of the intestinal barrier will be conducted using this cell line, which originates from nontumorigenic tissue. On the other hand, the Caco-2 cell line will be used in this experiment as a standard model of the small intestine in a mature organism, although it originates from tumorigenic tissue. Although in studies of the bioavailability of compounds, it is treated as a physiological healthy imitation of the intestinal barrier [33]. The Caco-2 cell line will be used in this system as a standard model of the small intestine of a mature organism. Cell lines will be exposed to optimal media fortified with the following: a) different doses of a mixture of aforementioned IRP to stimulate sensitization mechanisms, b) varying doses of IMCP for the monitoring of physiological/modulatory response, and c) IRP followed by IMCP to monitor the dominating mechanisms. As a positive control of inflammatory mechanism, cells will be treated with TNFα and LPS. The negative control will be the cell culture with no treatment, whereas the culture with the addition of solvent used for BPs (IRP/IMCP) suspension will be determined as a vehicle (VEH). The influence of BPs (IRP/IMCP) on cell proliferation/apoptosis/autophagy induction, cell cycle modulation, and the expression of genes of innate immune response and inflammatory markers (IL-1, IL-1R, IL-2, IL-4, IL-8, IL-10, TNFα, NFκB, and tight junction proteins) on mRNA and proteins level will be determined.

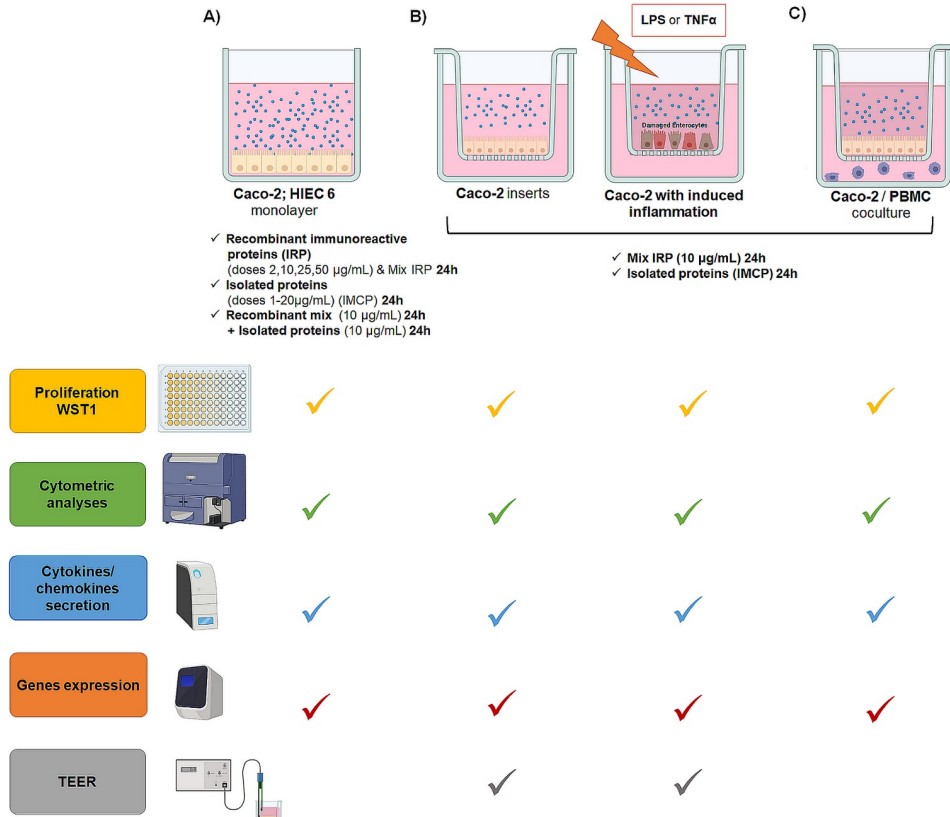

**Fig 2. Experimental scheme of *in vitro* study with the application of epithelial cell lines.**

## Assessment of immunoreactive properties of BPs in *in vivo* and *ex vivo* studies

The assessment of immunoreactive/immunomodulatory properties of BPs *in vivo* will be performed using a mouse model (BALB/cJRj) and *ex vivo* with cultures of induced mice lymphoid tissue (primary cultures). The experimental scheme of *in vivo* study is depicted in Fig 3A and *ex vivo* experiment in Fig 3B. In this study, BALB/cJRj mice will be three times immunized through intraperitoneal injection of the same mixture of IRP (tested *in vitro*) with adjuvant [34, 35]. After allowing one week of stabilization after the last immunization, animals will be intragastrically treated for 21 days with different variants of IMCP in a dose of 10 µg/mouse/ per day. These provided doses will be the equivalent of the daily consumption of one yogurt per human per day. The control (SHAM) group will be treated consistently with PBS. All procedures for animal study are summarized in the Table 2. We will evaluate humoral response (IgE/IgG/IgA content), blood cell phenotyping, and other morphology parameters. Additionally, we will assess the influence of treatments on the profile of mice intestinal microbiota and its activity will be also tested. In postmortem collected tissues (small intestine, spleen, and MLN), we will examine the expression of primary immune response markers. Following the termination of animals, primary cultures of mouse splenocytes/ lymph nodes lymphocytes will be prepared and cells will be cultured as shown in the layout for epithelial cells in Exp. 2. with the confirmed allergens (β-lactoglobulin/ ovalbumin/ IRP). However, an additional positive control, Con A, will be included as a mitogen. In lymphoid tissue, we will investigate the

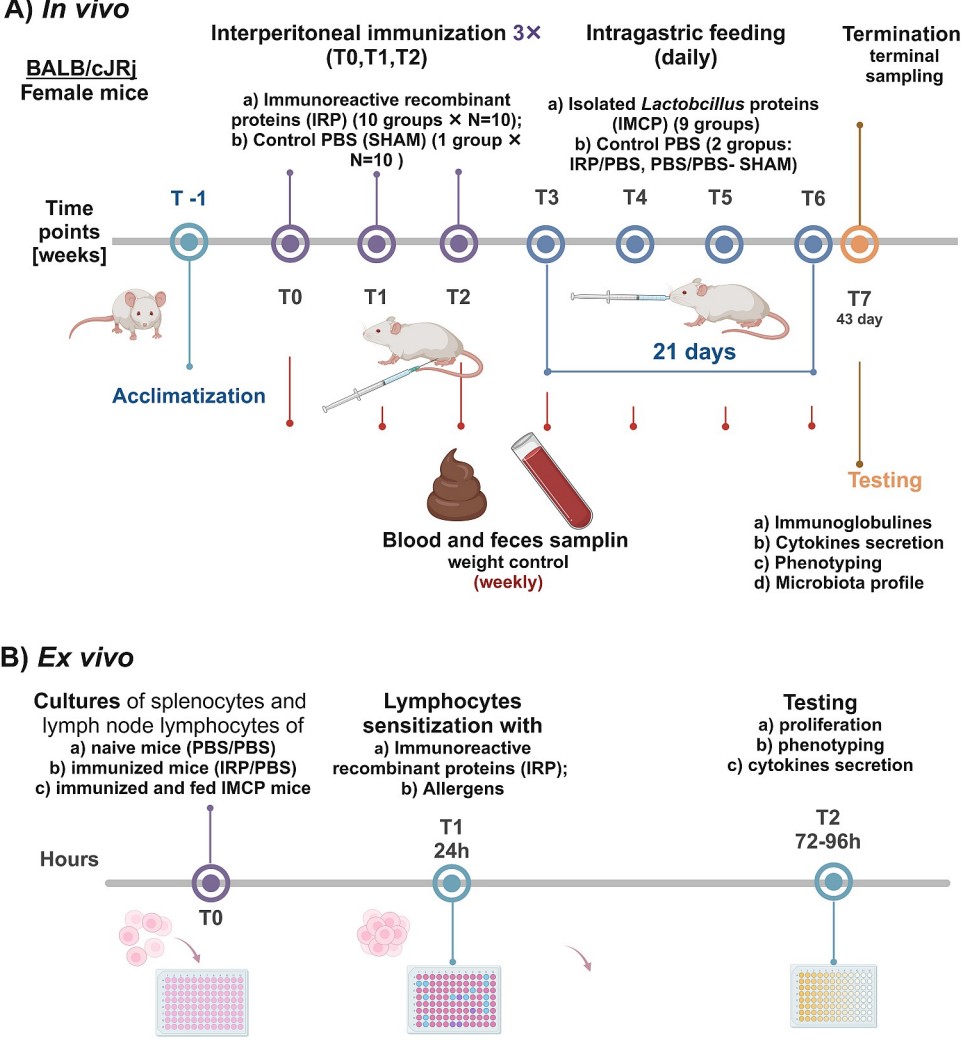

**Fig 3. Experimental scheme of *in vivo* study.** T-1—T7 refers to time points in the experiment.

impact of treatment on proliferation/cytotoxicity, cell cycle, and the effect on the expression of antigen uptake and presentation markers will be tested (i.e., on CD4,-8, -44, -c42,-1d1,-1d2,-209a, and 80).

## Material and methods

In the collected samples, the following laboratory analyses will be performed to verify the research hypotheses:

### Immunoreactivity determination

1. Bacterial cultures: Characterized *Lactobacillus* strains, from the Microorganisms Collection of the Institute of Animal Reproduction and Food Research (IAR&FR) [36], will be cultured in semi-anaerobic conditions at 37˚C for 18 h, with the pH being controlled in optimal MRS medium without fortification (CTRL) and with different lipids fortification (lipids solving will

**Table 2. Procedures for animal study.**

| No. | Name of the group | Intraperitoneal Immunization | Intragastrical Treatment | Number of animals per group [n] |
|---|---|---|---|---|
| 1 | CTRL (SHAM) | PBS | PBS | 10 |
| 2 | I/ CTRL (Immunized/CTRL) | IRP | PBS | 10 |
| | | | IMCP | |
| 3 | I/ MRS LcY | IRP | MRS *L. casei* LcY | 10 |
| 4 | I/ MRS 151 | IRP | MRS *L. bulgaricus* 151 | 10 |
| 5 | I/ MRS GG | IRP | MRS *L. rhamnosus* GG | 10 |
| 6 | I/ CM LcY | IRP | CM *L. casei* LcY | 10 |
| 7 | I/ CM 151 | IRP | CM *L. bulgaricus* 151 | 10 |
| 8 | I/ CM GG | IRP | CM *L. rhamnosus* GG | 10 |
| 9 | I/ BM LcY | IRP | BM *L. casei* LcY | 10 |
| 10 | I/ BM 151 | IRP | BM *L. bulgaricus* 151 | 10 |
| 11 | I/ BM GG | IRP | BM *L. rhamnosus* GG | 10 |

I, immunized; MRS, optimal Man, Rogosa, Sharpe medium with no modifications; CM, cow's milk-based lipid composition; BM, buttermilk-based lipid composition; IRP, Recombinant IgE-immunoreactive BPs; IMCP, Isolated *Lactobacillus sp.* proteins obtained from modified lipids cultures.

be enhanced with ultrasounds and mechanical emulsification). The number of bacteria will be controlled on agar MRS plates. The condition and activity of bacterial cells will be monitored with cytometric analysis [37] and bacterial gene expression analysis.

2. Lipids analysis: A modified Folch extraction method with SPE purification lipids from the media and its analysis with thin-layer chromatography/photochemical methods/ HPLC-ESI MS analyses according to Verardo et al. (2013) will be performed [28].

3. Lipids oxidation testing: It will be performed with the TBARS spectrophotometric analysis method adapted to dairy matrices [38], with the detection at 532 nm on the microplate reader Infinite M1000 PRO (Tecan).

4. BPs isolation: It will be performed from 50 mL of liquid cultures with a modified in-department method [26]. For *in vitro* and *in vivo* studies, BPs fractions will be dialyzed through the 15 kDa membrane and filtered.

5. BPs comparative proteomic analysis: The qualitative and quantitative analyses of proteins will be performed by external service with LCMS technique (IBB PAS, Warsaw) using a nanoAcquity ultra-performance LC (UPLC) system coupled to an LTQ-Orbitrap Velos mass spectrometer [37]. Spectra will be used for searching through the created Lactobacillus database by Mascot Server.

6. Visualization of the proteins: It will be carried out for 15 µg of BPs by SDS-PAGE with the Tricine method optimized for BPs separation using the Mini PROTEAN 3 Cell apparatus (Bio-Rad) [39].

7. Immunoblotting with human sera method: The determination of BPs immunoreactivity will be performed according to the protocols previously optimized in the department [26, 32, 37]. Briefly, 2-DE of 50 µg of BPs will be followed by wet electrotransfer, blocked, and incubated overnight (4°C) in a solution of human sera (agreement of Bioethics Committee No: 24/ 2022 from 27[th] of October 2022). Simultaneous detection of BPs (IRP) reacting with IgE and IMCP reacting with the mixture of IgG/A/M will be performed with two types of secondary antibodies labeled with, i.e., 680 and 800 nm fluorescent markers and visualization with the ChemiDoc Imaging Systems (Bio-Rad). Also SDS-PAGE with tricin separation of 10 µg of BPs will be followed by wet electrotransfer, blocked, and incubated overnight (4°C) in a solution of human sera. Detection of BPs reacting with IgE/IgG1/IgG4 will be performed with secondary

antibodies conjugated with horseradish peroxidase visualized with the ChemiDoc Imaging Systems (Bio-Rad).

8. MALDI-TOF MS/MS spots identification: Immunoreactive spots will be cut from the gels, destained, trypsinized, and finally desalted using ZipTip C18 tips and identified with MALDI-TOF MS/MS using an Autoflex Speed spectrometer and settings for Mascot Server like in LCMS [26].

## Impact of BPs on epithelial barrier integrity and physiology in *in vitro* study

9. Epithelial and lymphatic cell cultures: Cell lines will be purchased from ATCC collection and cultured at 37˚C with 95% humidity and 5% $CO_2$ in optimal media, following to the provided protocols. Caco-2 cell line and PBMC will be cultured in DMEM medium. The HIEC-6 cell line will be cultured in OptiMEM 1 with/without 10 ng/mL of EGF. RPMI-1640 medium will be used also for SPL and lymph nodes lymphocytes isolated from mice according to the protocol adapted in the department. Aliquots of the IRP and IMCP will be diluted in a particular medium to a final tested concentration of 1–50 and 1–20 μg/mL, respectively. For induction of controlled cytotoxic reaction in epithelial cells, TNF-α and LPS will be used. As a stimulation for lymphocytes Con A (10 μg/mL) will be used as a positive control. Cells with no treatment will be considered a negative control [33, 34].

10. Cell viability and proliferation testing: It will be performed with WST-1 and LDH spectrophotometric assays according to the manufacturer-provided protocol.

11. Cell cycle analysis: It will be evaluated by flow cytometry with lysis buffer containing PI and 1 mg/mL RNase A [40].

12. Apoptosis testing: It will be conducted using commercially available kits based on AnexinV-FITC with cytometry detection [40].

13. TEER: It will be measured with Millicell® ERS-2 Voltohmmeter (EVOM2).

14. mRNA expression of biomarkers: It will be analyzed using a two-step qPCR protocol. The reverse transcription step will be done with specific starters using Mastercycler® nexus System and real-time PCR reaction on QuantStudio 6 Pro Systems with the normalization to GAPDH and β-actin housekeeping genes [40].

15. Cytokines secretion: It will be tested using the Luminex 200 system, which allows for multiplexing based on ELISA kits designed for the simultaneous detection of even 46 different factors in low volumes (i.e., 25 μL of the sample).

## Assessment of immunoreactive properties of BPs in in vivo and ex vivo studies

16. 'Mice housing and experiment: 6 weeks old, female BALB/cJRj mice will be obtained from the JANVIER LABS, Le Genest-Saint-Isle, France and will be housed in individually ventilated cages with 12h day/night phase cycle at the Animal Facility of IAR&FR of PAS. All animals will be randomly distributed 5 animals per cage. Each group of animals (control and treated) will consist of $n$ = 10 animals. Water and a standard diet (TPF, Altromin, Germany; free of milk proteins and BSA) will be provided ad libitum. The mice will undergo acclimatization and handling before experiments for 2 weeks (T -1). After acclimatization mice will be immunized via intraperitoneal injections 200 μL of IRP (100 μg/mL of PBS) with aluminum adjuvant (1:1 v/v). The procedure will be repeated three times at weekly intervals (T0, T1, T2). The control group will be intraperitoneally injected with 200 μL PBS. After the last immunization and one-week stabilization (T3) mice will be intragastrically treated with particular tested IMCP (in a dose 10 μg /mouse/day) for the next 21 days (T3-T6). The control group will be

intragastrically treated with 200 μL PBS. Afterward, the animals will be terminated (T7), and tissue samples will be immediately collected in the 43$^{rd}$ day of the experiment counted from T0 to T7 (Fig 3A). All these protocols have been previously implemented in our department and received approval by the Ethics Committee of the University of Warmia and Mazury (Olsztyn, Poland), (agreement No: 16/2023 from 15$^{th}$ of February 2023).'

17. Lymphocyte phenotyping: Peripheral blood lymphocytes will be phenotyped using BD fluorescent-labeled antibodies to CD4, CD25, and CD8a FoxP3 a.o. [34, 35].

18. Serum IgE, IgG, IgA, and cytokines: The levels will be determined using ELISA kits for Luminex 200. Tests will be performed according to the manufacturer's protocol.

19. Quantification of Fecal Microbiota by qPCR: Microbiota profiling will be conducted according to the previously established qPCR protocol. It will be quantitative analysis using genera- and group-specific 16S rRNA gene primers for qPCR. Content of the cecum will be collected post-mortem for the microbiota analysis. Fresh samples will be immediately frozen in sterile tubes in liquid nitrogen. This analysis will target major bacterial groups (Bacteroides-Prevotella-Porphyromonas, Clostridium cluster IV represented by *Clostridium leptum*, Blautia represented by *Clostridium coccoides*) and genera (*Atopobium*, *Enterococcus*, *Bifidobacterium* and *Lactobacillus*) using the QuantStudio 6 System. The results will be expressed as log10 of the cell number per gram sample wet weight. A detailed description of the procedure has been presented in previous studies [34, 41].

## Statistical analysis plan

The data obtained from this complex *in vitro*, *in vivo*, *ex vivo*, and *in silico* experiments are amenable to the analysis using standard statistical techniques. These analyses will be carried out using STATISTICA version 13.3 (StatSoft, Tulsa, USA), GraphPad Prism, and supplemented with online *in silico*, based on artificial neural network, tools will be applied according to our previous experience. Experiments 1 and 2 will be conducted in triplicates, and Experiment 3, involving animal use, will be limited to duplicate repetitions. The number of animals was calculated based on the Test power analysis of STATISTICA13.3 (StatSoft, USA), based on the standard deviation of historical data from previous experiments and assuming a 5-point significance test in the analysis. Based on that test and due to the parallel *in vitro* and *ex vivo* tests calculated size of each group will be *n* = 10 for experiment.

## Ethical aspects

In this work, the biological material from the previously created sera bank (sera of patients with poli-allergy/and healthy individuals control) collected in the previous project (reg. no. N N312 311939) will be used. No additional sera will be procured for this research. For bank development, the commission acceptance (No2/2010) was obtained by the Institute of Animal Reproduction and Food Research of the Polish Academy of Sciences in Olsztyn from the Bioethics Committee of the Faculty of Medical Sciences of the University of Warmia and Mazury in Olsztyn. Although the biological material has been previously collected and characterized, the planned procedures necessitate the commission's approval for the reuse of sera. The experimental design and procedures have been duly endorsed by the Bioethics Committee of the Faculty of Medical Sciences of the University of Warmia and Mazury in Olsztyn (agreement No: 24/2022 from 27$^{th}$ of October 2022). The planned methodologies offer the advantage of substituting invasive tests involving patients using normalized *in vitro* immunoreactivity analysis methods. These methodologies will rely on well-established and widely used immunoblotting methods using human sera, thus minimizing the risk to the patient's health. In addition, commercially available, established cell lines, originating from human hosts and obtained

from international collections, will be used to eliminate the need for human biopsy analysis during research. The use of commercially available human epithelial cell lines facilitates observations related to the interaction of BPs with mucosa by omitting *in vivo* invasive tests. These cellular models align with global food safety research agendas in our project. Working with collected human-derived biological material contributes to cost reduction and reduces the need for research conducted using a large number of animals in a model. At the same time, it allows us to maintain a high standard of research and reproducibility.

Commercially available human and mouse immune cell lines will be used to reduce the scale of research in the mouse model. Mouse immune cell lines will be used directly to assess the immunomodulatory effects of BPs. Through *in vitro* and *ex vivo* tests on isolated lymphocytes, we will aim to replace multiple treatment options conducted on living organisms. Modern and optimized multiplex-based techniques will enhance the refinement of the experimental part. The experimental design and procedures for the mice model have been approved by the Ethics Committee of the University of Warmia and Mazury (Olsztyn, Poland), (agreement No: 16/2023 from 15[th] of February 2023). The animal *in vivo* experiment will be limited only to nine groups treated with BPs and two control groups. An abundance of groups ($n = 10$) and numerous control groups as well as a refinement of *in vivo* studies through parallel planned *in vitro* and *ex vivo* tests and the use of modern and sensitive methods of multiplexing allow us to limit the range of *in vivo* tests to a double repetition. Group size ($n = 10$) is dictated by the requirement to isolate lymphocytes for *ex vivo* testing.

## Data management plan

To safeguard the privacy of serum providers and ensure confidentiality, all personal data are stored in password-protected files, and safeguarded against unauthorized access by third parties. Access to raw data and materials is restricted to project team members only. In publications, only completely anonymized data, such as mean values and group statistics, will be made available in data repositories. Access to (nonsensitive) data will be granted upon individual request to the PI and data manager representing IAR&FR of PAS. Specific data (proteomic and genetic) will be deposited in in specialized databases like PRIDE and NCBI. Data will be deposited in general repositories like RepOD, which accept multiple data types.

## Discussion

This comprehensive *in vitro*, *in vivo*, and *ex vivo* study was designed to evaluate the impact of BPs on immune parameters in both mature and immature epithelial barriers and female immunized mice in the context of hypersensitivity/tolerance development. Allergy is a prevalent, serious, and costly condition, necessitating intervention to mitigate its distressing symptoms. In the therapy and prophylaxes of hypersensitivity, probiotics, symbiotics, and special fermented formulas are extremely investigated. Bacterial structures and metabolites like exopolysaccharides, volatile fatty acids, and nucleic acids manifest immunomodulatory effects and induce immunoregulatory mechanisms [7, 17]. However, the role of microbial proteins in the bacterial structure is of particular significance, especially considering the consumption of reported dairy beverages and probiotic preparations. In some instances, the consumption of probiotic preparations has been associated with a decline in the health status of individuals suffering from food allergies [15, 16]. Instead of the conventional IgG4-mediated and IgA-mediated tolerance pathways, heightened IgE-mediated and IgG1-mediated pathways characteristic of allergies were observed in these subjects. Although it can be assumed that impairment in the proper functioning of the regulatory axis (MyD88/RORγt) may be responsible for this phenomenon [15, 20], it has not been verified in terms of reaction to proteins of potentially

health-promoting strains. Furthermore, the regulatory role of lipids on BPs expression has not been explored. It is conceivable that the addition of lipids to the microbial medium can significantly impact the fluidity and, consequently, the permeability of the bacterial cell membrane, influencing the profile of the expressed and exposed microbial proteins. This aspect is also important in the era of the popular keto diet, where the presence of lipids in the diet is crucial. In this context, the common practice of exposing the organism with BPs of novel foods, particularly during the stage when diseases such as allergies and intolerance tend to develop (i.e., in infancy and childhood) places an additional burden on the overwhelmed, still-developing immune system. There is a notable absence of studies examining isolated BPs in terms of their sensitizing potential. To investigate this knowledge gap, we have outlined a study protocol of a comprehensive experiment that investigates the effects of BPs in *in vitro* protocol imitating the epithelial barrier, validated in a mice model, with specific emphasis on inflammation, apoptosis, autophagy, and intestinal barrier integrity.

## Strengths of the planned study

To date, the majority of studies with LAB have been conducted using live bacteria or thermally inactivated bacteria, enabling the analysis of the impact of a combination of bacterial components and not the BPs themselves. One of the primary strengths of this study lies in its comprehensive approach, where the effects of purified and recombinant BPs will be rigorously examined through a series of *in vitro*, *in vivo*, *ex vivo*, and *in silico* tests. Moreover, in the *in vitro* trial, diverse cell lines with different physiological states (inflamed and physiological), different origins (healthy and cancer origins), as well as with various stages of maturity will be tested. Hence, many experimental variants in *in vivo* and clinical trials would be logistically challenging. Another notable strength is that an *in vitro* study will be supported with an *in vivo* trial on a mouse model. The trial has been meticulously planned with a statistically determined sample size ($n = 10$ mice/group) to ensure that the results obtained are statistically significant. The use of a female-only model eliminates gender as a potential confounding factor, enhancing the homogeneity of the results. The collected samples will undergo a wide range of analyses, providing valuable insights to better understand the physiological effects of BPs.

## Potential risks

The results of serum-based, immunoreactivity *in vitro* tests of BPs are subject to variability due to the activity of human sera. To ensure proper repeatability and sensitivity of the method, we have previously determined the minimum number of human sera required and the optimal amount of total IgE. Within the IAR&FR sera collection, we have access to more than 500 characterized and described [30, 31] allergic sera, which are aliquoted and readily available if replacements are needed. Another potential limitation is the low volume of medium available after cell culturing, as well as the limited serum volume from individual mice, considering the planned number of analyses, but this limitation will be mitigated by using multiplex kits designed for various human and mice cytokines and markers (analysis in 50 µL medium and serum). In the *in vivo*/*ex vivo* trial, the number of isolated lymphocytes from MLN and other peripheral lymph nodes could be a limiting factor. To address this challenge and ensure robust results, we will increase the number of animals per group from 8 to 10, thereby obtaining a sufficient number of cells for *ex vivo* testing.

## Limitations

Our study is limited to adult female mice of conventional breeding. Future trials could explore other age groups, including immature mice, as well as primary immune response gene

knockout mice, to further broaden our understanding of these effects. It would be also interesting to investigate whether the effects observed in *in vitro* and *in vivo* trials align with those in the interventional nutritional study involving human participants.

## Implications

Despite the aforementioned limitations, we believe that the proposed trial will significantly advance our understanding of how the *Lactobacillus* microbial proteins impact epithelial barrier permeability, immune system functioning, and oversensitivity *vs* tolerance development.

## Conclusion

In this article, we have presented a rationale and study protocol for *in vitro*, *in vivo*, and *ex vivo* trials to better understand the immunomodulatory/immunoreactive and physiological effects of microbial proteins of *Lactobacillus* strains. We will specifically examined the impact of lipids on the immunoreactivity of bacterial proteins within the context of their role in allergy and oversensitivity development. The results generated from this research endeavor hold the potential to shed new light on the intricate interactions between lactic acid bacteria and the human immune system. Furthermore, the meticulous selection of a raw dairy material with an optimal lipid composition has the potential to enhance nutraceutical properties and the safety of products containing these bacteria. The novel insights gained from this study may serve as a foundation for the implementation of "immunoreactivity of bacteria" analysis as a novel discriminant in food safety assessments.

## Acknowledgments

We appreciate the skillful assistance of Translmed Publishing Group. The authors thank the patient of Allergica and Atarax medical centres in Olsztyn (Poland) and The Voivodeship Rehabilitation Hospital for Children in Ameryka (Poland) and their families which participated in tests.

## Author Contributions

**Conceptualization:** Anna Maria Ogrodowczyk.

**Data curation:** Anna Maria Ogrodowczyk, Ewa Romaszko.

**Funding acquisition:** Anna Maria Ogrodowczyk.

**Investigation:** Anna Maria Ogrodowczyk.

**Methodology:** Anna Maria Ogrodowczyk.

**Project administration:** Anna Maria Ogrodowczyk.

**Resources:** Anna Maria Ogrodowczyk.

**Visualization:** Anna Maria Ogrodowczyk.

**Writing – original draft:** Anna Maria Ogrodowczyk.

**Writing – review & editing:** Anna Maria Ogrodowczyk, Ewa Romaszko.

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
