## [Decision Letter · Decision Letter 0]

28 Feb 2024

PONE-D-23-30103Study protocol: The role of milk matrix lipids in programming the immunoreactivity of proteins derived from lactic acid bacteriaPLOS ONE

Dear Dr. Ogrodowczyk,

Thank you for submitting your manuscript to PLOS ONE. After careful consideration, we feel that it has merit but does not fully meet PLOS ONE’s publication criteria as it currently stands. Therefore, we invite you to submit a revised version of the manuscript that addresses the points raised during the review process. Please consider the suggestions done by both reviewers

We look forward to receiving your revised manuscript.

Kind regards,

Guadalupe Virginia Nevárez-Moorillón, Ph.D.

Academic Editor

PLOS ONE

Journal Requirements:

Reviewers' comments:

Reviewer's Responses to Questions

**Comments to the Author**

1. Does the manuscript provide a valid rationale for the proposed study, with clearly identified and justified research questions?

Reviewer #1: Yes

Reviewer #2: Yes

2. Is the protocol technically sound and planned in a manner that will lead to a meaningful outcome and allow testing the stated hypotheses?

Reviewer #1: Partly

Reviewer #2: Yes

3. Is the methodology feasible and described in sufficient detail to allow the work to be replicable?

Reviewer #1: No

Reviewer #2: Yes

4. Have the authors described where all data underlying the findings will be made available when the study is complete?

Reviewer #1: Yes

Reviewer #2: Yes

5. Is the manuscript presented in an intelligible fashion and written in standard English?

Reviewer #1: Yes

Reviewer #2: Yes

6. Review Comments to the Author

You may also provide optional suggestions and comments to authors that they might find helpful in planning their study.

Reviewer #1: This study protocol resorts to a multidisciplinary approach and several in vitro and in vivo models to evaluate LAB microbial proteins on inflammation, apoptosis, autophagy and intestinal barrier integrity. The role of milk matrix lipids as determinant in the development of immune responses to microbial proteins is also envisioned in the project plan. However, there are major criticism, which must be considered before publication. Here is a list of major points:

One of the major flaws concerns the mice study. At point 16 of the “Laboratory Analysis” section, the procedures involving mice are only scanty described. There are no indications concerning the modality of housing mice (i.e if animals will be put on a standard diet, the daily dark/light cycle, the animal house temperature, the number of animals/cage etc…). There are no indications on the age of the animals, their weight and the source of the strain. Furthermore, only female will be used. I disagree with the last sentence of the paragraphs describing the “Strengths of the study”, since both sexes must be considered in order to get appropriate and more complete sex-driven information in the responses under study as indicated by several regulatory agencies and scientific journals (see: Clayton and Collins, Nature 509, 282 2014; Br J Pharmacol 176, 4081-4086, 2019). Another point of criticism is the number of animals used for both in vivo immunization and to evaluate the response with IMCP. Apparently 10 and 9 animals will be used in the treatment group and just 5 in the control groups. This is quite odd since usually the number of CTR and treated animals should be the same to obtain a correct statistical analysis. The power analysis used to predict these number of animals in the study has not been described. The term group and number of animals is often erroneously interchanged, see as an example figure 3 :10 groups, 9 groups….do the Author mean 10 animals?

How will be fecal samples collected for the microbiome analysis?

I also have concern regarding point 19 describing the microbiome analysis. There are no indications on the sequences which will be amplified, and the PCR is very poorly described. The method is per se questionable since NGS analysis of the 16 bacterial RNA would give more complete information on the possible changes in the composition and abundance of the main Phyla and Genera. It is not clear why some strains (not groups as indicated by the Author) have been chosen (like Clostridium coccoides and Clostridium leptum…are they linked with changes in immune response in the enteric microenvironment?). There is a bit of confusion when distinguishing strains and genera: Bacteriodes-Prevotella-Porphyromonas are not strains but genera.

I would also change the structure of the text:

The Introduction section and “Research objectives and hypothesis” are a bit long and should be shortened and merged;

At page 3 of the introduction the word “applicant” should be substituted with author, since this is not a grant application;

Use abbreviations consistently once explained the first time in the text;

Ethical aspects should follow the Statistical Analyses

I suggest to write a Material and Methods section subdivided into: Experimental design; First Tier (immunoreactivity determination with the relevant laboratory analyses: from point 1 to 8); Second Tier (assessment of immunoreactive properties…. with the relevant laboratory analyses: from point 9 to 15); Third Tier (impact of BPs on epithelial barrier… with the relevant laboratory analyses : from point 10 to 19

Reviewer #2: In this study protocol, the authors present a proposal to evaluate the effects of proteins produced by bacteria in intestinal tissue and their effect on the immune system specifically in hypersensitivity reactions. The protocol includes in vitro, in vivo, and ex vivo assays well planned and conducted with solid and feasible methodological proposals; the corresponding authorization of the bioethics committee was considered: additionally, important situations such as potential risks and limitations are described.

For the publication of the protocol I only suggest improving the quality of the figures.

7. PLOS authors have the option to publish the peer review history of their article (what does this mean?). If published, this will include your full peer review and any attached files.

Reviewer #1: No

Reviewer #2: No

---

## [Author Response · Author response to Decision Letter 0]

14 Mar 2024

Reviewer #1: 

The authors would like to thank the reviewers for their insightful and constructive review of this work. Reading the review, it appears that the in vivo experiment was not described clearly enough, so the description of protocols was supplemented with issues that were questionable to the reviewer. We would like to mention that the compliance of all questionable issues can also be verified in the S3 and S4 applications attached as supplements to this work. There are applications submitted to the ethics committee along with the consent granted by the ethics committee to conduct the study. 

Unfortunately, the authors would like to emphasize that at this stage it is not possible to modify the microbiota analysis method because the budget of the granted grant is a limiting factor. Nevertheless as far as we are concerned all our planned methods are still valid and NGS is not the only method we can get informative results. This description concerns funding that has already been granted and all methods were reviewed by 5 independent reviewers, and a 10-person ethical committee for animal research, so we hope we planned everything properly except we did not describe it so well here and that is why there are doubts. We will try to explain as clearly as possible why such a set of tests was selected in the planned in vivo study. We will also try to provide as much evidence as possible about the repeatability of our results in previous studies using the selected methods.

At the same time, we also wanted to explain that some of the data was organized in this way because we focused mainly on explaining the complex and multi-faceted study design and the methodology, especially the one that had already been published by our team and was not modified in this project, was only quoted as a reference. We will try to fill in the aspects that seem missing.

We will address the provided comments: 

1. One of the major flaws concerns the mice study. At point 16 of the “Laboratory Analysis” section, the procedures involving mice are only scanty described. There are no indications concerning the modality of housing mice (i.e if animals will be put on a standard diet, the daily dark/light cycle, the animal house temperature, the number of animals/cage etc…). 

Thank you for this comment. We added the section regarding mice hoteling in the methodology section as below.

‘Mice housing and experiment– 6 weeks old, female BALB/cJRj mice will be obtained from the JANVIER LABS, Le Genest-Saint-Isle, France and will be housed in individually ventilated cages with 12h day/night phase cycle at the Animal Facility of IAR&FR of PAS. All animals will be randomly distributed 5 animals per cage. Each group of animals (control and treated) will consist of N=10 animals. Water and a standard diet (TPF, Altromin, Germany; free of milk proteins and BSA) will be provided ad libitum. The mice will undergo acclimatization and handling before experiments for 2 weeks (T -1). After acclimatization mice will be immunized via intraperitoneal injections 200 μL of IRP (100 μg/mL of PBS) with aluminum adjuvant (1:1 v/v). The procedure will be repeated three times at weekly intervals (T0, T1, T2). The control group will be intraperitoneally injected with 200 μL PBS. After the last immunization and one-week stabilization (T3) mice will be intragastrically treated with particular tested IMCP (in a dose 10 µg /mouse/day) for the next 21 days (T3-T6). The control group will be intragastrically treated with 200 μL PBS. Afterward, the animals will be terminated (T7), and tissue samples will be immediately collected in the 43rd day of the experiment counted from T0 to T7 (Figure 3A). All these protocols have been previously implemented in our department [29] and received approval by the Ethics Committee of the University of Warmia and Mazury (Olsztyn, Poland).’ 

2. There are no indications on the source of the strain. 

Thank you for the comment. Some additional data about the source of strain we put here.

At the time of writing the project and this manuscript, it was not obvious from which source the mice strain would be available to us at the time of research because a lot depends on public procurement in the institution. We do not decide about tender results. Therefore, only the mouse strain is given in the previous manuscript and not its source. Now we know it will be JANVIER LABS BALB/cJRj Strain. So we filled in the information that was missing.

The strain was selected by MacDowell from a stock of outbred albino mice and then transferred to Snell (The Jackson Laboratory) at F26 in 1935. The “c” was added to the strain name by Dr Snell to indicate that the genotype for the color locus is c/c, hence the name BALB/c. 1935: From Dr Snell to Drs Heston and Andervont (NIH), separation of the BALB/cJ and BALB/cByJ strains at the F38 generation, when some of Dr. Snell’s BALB/cSn mice were transferred to Drs Andervont and Heston. 1974: Breeders were transferred to the Production Department of The Jackson Laboratory at F136 and the J was added.

3. Furthermore, only female will be used. I disagree with the last sentence of the paragraphs describing the “Strengths of the study”, since both sexes must be considered in order to get appropriate and more complete sex-driven information in the responses under study as indicated by several regulatory agencies and scientific journals (see: Clayton and Collins, Nature 509, 282 2014; Br J Pharmacol 176, 4081-4086, 2019). 

Thank you for your vigilance and attention. Indeed, such a formulation of the ‘Strenghts’ chapter raises doubts.

We mean that using one gender eliminates the phenomenon of gender variability in this work. We do not insist on the universality of the results for both sex. We mean the repeatability of the results in the research setup planned like here. Despite being aware of the fact that gender variability plays an important role in the experiments in general but female model was considered for a long time as the most consistent. Here, we consciously and deliberately gave male units up which we explain below. 

However, to avoid any doubts, we decided to remove word ‘reliability’ from this sentence localized in the ‘strengths’ chapter.

For the planned experiment we use female mice which, according to previously published research and our own observations, in microbiological tests are capable of a higher production of MLN cells, which we are particularly interested in (that tissue is extremely important for us and cells are limited), and a higher percentage of CD4+/CD8+ cells in spleens compared to male mice of this breed are also referred [Elderman, M.,et al.(2018). https://doi.org/10.1186/s13293-018-0186-6 ]. Both parameters are crucial in planned tests. Females are also calmer and less often fight for dominance than males, so they are less likely to be behaviorally affected by the stress associated with a dominant individual. In our experience, we have to catch females from the group less often than males, which generates fewer losses in individuals. In the era of applying the 3R principle (Replacement, Reduction, Refinement), in animal research, we focused on female animals because it allows us to obtain the necessary tissue in the quantity necessary for research. We would have to use more males. 

Since these are the first studies in this series and there will be 11 groups of subjects and each group will consist of 10 individuals (there will also be a pilot-dose testing study), it is still a large experiment taking into account the scope of the planned expertise. Moreover, the project is planned to be implemented only for 3 years, and since it is not possible to employ additional people under the grant, extending the research to include an equal group of males would be impossible from the point of view of research implementation. For the funds' providers, it would be unreliable in terms of feasibility.

However, we do not deny that we have further plans to continue in vivo research. It is planned, among others, to apply for projects allowing to determine the role of recombinants/isolated bacterial proteins on immunodeficient mice. So we will definitely take this valuable comment into account in further research.

4. Another point of criticism is the number of animals used for both in vivo immunization and to evaluate the response with IMCP. Apparently 10 and 9 animals will be used in the treatment group and just 5 in the control groups. This is quite odd since usually the number of CTR and treated animals should be the same to obtain a correct statistical analysis. The power analysis used to predict these number of animals in the study has not been described. The term group and number of animals is often erroneously interchanged, see as an example figure 3 :10 groups, 9 groups….do the Author mean 10 animals?

We are really sorry. There must have been some misunderstanding. Throughout the work and in the applications to the ethics committee, we emphasize that all groups will be equal and will consist of 10 individuals. 5 animals in in cage but the experiment will be in duplicate. So N of each group is 10. It was mentioned in the strengths and in the statistical description about duplicates. In Figure 3 we describe the number of immunized groups (10 groups will be immunized with IRP). 9 groups will be fed with particular IMCP. There will be two CONTROL groups. One PBS/PBS (SHAM) and one IRP/PBS (Control of immunization) will be treated. We will use a total of 110 animals for the experiment. 11 groups x 10 animals each. We do not interchange words for group and number of animals but to better understand we prepared the table with procedures, treatments, and number of animals. We understand it might have been confusing. We are sorry for that. In Figure 3 we used also T-1 – T7 which refers to time points in the experiment. We added the description to make everything less confusing. 

We corrected the statistical description in the manuscript too. We included that information only in the ethic committee application but we added that to the manuscript too to make it more transparent. Unfortunately, we do not know what is the type of test that STATSOFT 13 uses for the power estimation. We know that we performed it according to the manufacturer's instructions. We included that description to the manuscript. 

‘The Test power analysis of STATISTICA13.3 (StatSoft, USA), based on the standard deviation of historical data from previous experiments and assuming a 5-point significance test in the analysis was applied to assess the size of groups.’

The table 2 was added as below

Table 2. Procedures of animal study

No. Name of the group Intraperitoneal Immunization Intragastrical Treatment N of animals

per group

1 CTRL (SHAM) PBS PBS 10

2 I/ CTRL (Immunized/CTRL) IRP PBS 10

 with IMCP 

3 I/ MRS LcY IRP MRS L. casei LcY 10

4 I/ MRS 151 IRP MRS L. bulgaricus 151 10

5 I/ MRS GG IRP MRS L. rhamnosus GG 10

6 I/ CM LcY IRP CM L. casei LcY 10

7 I/ CM 151 IRP CM L. bulgaricus 151 10

8 I/ CM GG IRP CM L. rhamnosus GG 10

9 I/ BM LcY IRP BM L. casei LcY 10

10 I/ BM 151 IRP BM L. bulgaricus 151 10

11 I/ BM GG IRP BM L. rhamnosus GG 10

I, immunized; MRS, optimal Man, Rogosa, Sharpe medium with no modifications; CM, cow’s milk-based lipid composition; BM, buttermilk-based lipid composition; IRP, Recombinant IgE-immunoreactive BPs; IMCP, Isolated Lactobacillus sp. proteins obtained from modified lipids cultures.

5. How will be fecal samples collected for the microbiome analysis?

Content of the cecum will be collected post-mortem for the microbiota and SCFA analysis but fecal samples will be collected weekly for SCFA and sIgA analysis. Fresh stool samples and cecum contents for analysis will be immediately frozen in sterile tubes in liquid nitrogen. 

6. I also have concern regarding point 19 describing the microbiome analysis. There are no indications on the sequences which will be amplified, and the PCR is very poorly described. The method is per se questionable since NGS analysis of the 16 bacterial RNA would give more complete information on the possible changes in the composition and abundance of the main Phyla and Genera. It is not clear why some strains (not groups as indicated by the Author) have been chosen (like Clostridium coccoides and Clostridium leptum…are they linked with changes in immune response in the enteric microenvironment?). There is a bit of confusion when distinguishing strains and genera: Bacteriodes-Prevotella-Porphyromonas are not strains but genera.

We are sorry that the point 19. raises concerns. In this publication, we refer to manuscripts in which both the description of the analyses and the preparation of the standard curve for qualitative analysis are precisely described. We do not modify the protocol that is why we only briefly mentioned about planned analysis and added references and omitted the description of the method which is long. For the reviewer's request we will upload here the protocol in the response. We planned quantitative analysis using genera- and group-specific 16S rRNA gene primers for qPCR despite knowing that NGS analysis would be more informative. The use of the most modern method, although we know that it is more informative, unfortunately, might be not available for financial reasons. If any company offers us a reasonable price for analysis in a tender, we will accept it. However, to conduct microbiota testing cost-effectively, we planned an expression analysis method. Nevertheless as far as we are concerned our planned method is still valid and NGS is not the only method we can get informative results.

The description of the protocol

Real-Time PCR Quantification of Intestinal Microbiota.

Construction of a Standard Curve. 

To construct a standard curve for the quantification of particular microbiota populations, we will use strains representing the Bacteroides-Prevotella-Porphyromonas group, the Clostridium group IV (representing Clostridium leptum), Blautia group (representing Clostridium coccoides), Bifidobacterium, Enterococcus, and Lactobacillus as the main groups of intestinal microbiota. We will apply the approach described by Vahjen et al. (2007), with modifications. Strains which will be used originate from the German Microorganism and Cell-Culture Collection (Leibniz-Institut DSMZ-Deutsche Sammlung von Mikroorganismen und Zellkulturen GmbH, Braunschweig, Germany) and the Culture Collection of the Institute of Animal Reproduction and Food Research (Polish Academy of Sciences, Olsztyn, Poland). The bacterial cultures will be cultivated separately in appropriate conditions (details available upon request). The cell number of each culture will be determined using 4c,6-diamidino-phenylindole (Świątecka et al., 2013). Two milliliters of each culture will be centrifuged (5 min, 10,000 × g), washed with sterile PBS (pH 7.4), and centrifuged again. Cell pellets will be combined and mixed with 0.1 g of autoclaved cecal contents (autoclaved twice; 121°C, 15 min), and bacterial DNA will be isolated using the same method as for the cecal samples. Isolated DNA will be serially diluted 10-fold, and dilutions from 10−1 to 10−7 will be used to construct the standard curves. We will carry out real-time PCR amplifications with genus- and groups-specific primers to obtain the curve for each of the tested bacterial populations. 

Real-time amplifications will be performed in QuantStudio 6 Flex system (Thermo Fisher, Life Technologies, Warsaw, Poland) in a total volume of 25 µL, consisting of 12.5 µL of SYBR Green JumpStart Taq ReadyMix (Sigma, Poznan, Poland), 200 µM of each primer, 1 µL of DNA diluted 10-fold, and PCR-grade water (Sigma). The primer sequences and annealing temperatures are listed in the table added to the revision. 

The temperature program includes 1 cycle of 95°C for 3 min and 35 cycles of 95°C for 20 s, the temperature of primer annealing (Table) for 30 s, and 72°C for 30 s with signal acquisition. After completion of the amplifications, we will prepare a melting curve to confirm the specificity of the PCR products. We will normalize the obtained values according to the dilution and weight of the samp

---

## [Editor Report · Decision Letter 1]

18 Mar 2024

Study protocol: The role of milk matrix lipids in programming the immunoreactivity of proteins derived from lactic acid bacteria

PONE-D-23-30103R1

Dear Dr. Ogrodowczyk,

We’re pleased to inform you that your manuscript has been judged scientifically suitable for publication and will be formally accepted for publication once it meets all outstanding technical requirements.

Kind regards,

Guadalupe Virginia Nevárez-Moorillón, Ph.D.

Academic Editor

PLOS ONE
---

## [Editor Report · Acceptance letter]

8 May 2024

PONE-D-23-30103R1 

PLOS ONE

Dear Dr. Ogrodowczyk, 

I'm pleased to inform you that your manuscript has been deemed suitable for publication in PLOS ONE. Congratulations! Your manuscript is now being handed over to our production team.

Kind regards, 

on behalf of

Dr. Guadalupe Virginia Nevárez-Moorillón 

Academic Editor

PLOS ONE